# *Candida* Worsens *Klebsiella pneumoniae* Induced-Sepsis in a Mouse Model with Low Dose Dextran Sulfate Solution through Gut Dysbiosis and Enhanced Inflammation

**DOI:** 10.3390/ijms23137050

**Published:** 2022-06-24

**Authors:** Wimonrat Panpetch, Pornpimol Phuengmaung, Pratsanee Hiengrach, Jiraphorn Issara-Amphorn, Thanya Cheibchalard, Naraporn Somboonna, Somying Tumwasorn, Asada Leelahavanichkul

**Affiliations:** 1Department of Microbiology, Faculty of Medicine, Chulalongkorn University, Bangkok 10330, Thailand; mon-med@hotmail.com (W.P.); pphuengmaung@gmail.com (P.P.); pratsaneeh@gmail.com (P.H.); 2Center of Excellence in Translational Research on Immunology and Immune-Mediated Diseases (CETRII), Department of Microbiology, Faculty of Medicine, Bangkok 10330, Thailand; jiraphorn298@gmail.com; 3Program in Biotechnology, Faculty of Science, Chulalongkorn University, Bangkok 10330, Thailand; thanya-_-@hotmail.com; 4Department of Microbiology, Faculty of Science, Chulalongkorn University, Bangkok 10330, Thailand; naraporn.s@chula.ac.th; 5Microbiome Research Unit for Probiotics in Food and Cosmetics, Chulalongkorn University, Bangkok 10330, Thailand

**Keywords:** *Candida* administration, gut leakage, dysbiosis, dysbiosis, inflammation, inflammatory cytokine, *Klebsiella pneumoniae*, microbiome

## Abstract

*Klebsiella pneumoniae* is an opportunistic pathogen and a commensal organism that is possibly enhanced in several conditions with gut dysbiosis, and frequently detectable together with *Candida* overgrowth. Here, *K. pneumoniae* with or without *Candida albicans* was daily orally administered for 3 months in 0.8% dextran sulfate solution-induced mucositis mice and also tested *in vitro*. As such, *Candida* worsened *Klebsiella-*DSS-colitis as demonstrated by mortality, leaky gut (FITC-dextran assay, bacteremia, endotoxemia, and serum beta-glucan), gut dysbiosis (increased Deferribacteres from fecal microbiome analysis), liver pathology (histopathology), liver apoptosis (activated caspase 3), and cytokines (in serum and in the internal organs) when compared with *Klebsiella-*administered DSS mice. The combination of heat-killed *Candida* plus *Klebsiella* mildly facilitated inflammation in enterocytes (Caco-2), hepatocytes (HepG2), and THP-1-derived macrophages as indicated by supernatant cytokines or the gene expression. The addition of heat-killed *Candida* into *Klebsiella* preparations upregulated *TLR-2*, reduced *Occludin* (an intestinal tight junction molecule), and worsened enterocyte integrity (transepithelial electrical resistance) in Caco-2 and enhanced *casp8* and *casp9* (apoptosis genes) in HepG2 when compared with heat-killed *Klebsiella* alone. In conclusion, *Candida* enhanced enterocyte inflammation (partly through *TLR-2* upregulation and gut dysbiosis) that induced gut translocation of endotoxin and beta-glucan causing hyper-inflammatory responses, especially in hepatocytes and macrophages.

## 1. Introduction

*Klebsiella pneumoniae*, a Gram-negative encapsulated aerobic bacillus in the Enterobacteriaceae family, is normal flora in the mouth, oropharynx, gastrointestinal (GI) tracts, skin, and intestines of the human hosts [1] with high virulence and antibiotic resistance [2] in nosocomial or hospital-acquired infections [3]. As such, 3% to 8% of all nosocomial bacterial infections in the United States are caused by *K. pneumoniae* [2]. The *Klebsiella* polysaccharide capsule protects the bacteria from opsonization, phagocytosis, and bactericidal activity by the host [4], while lipopolysaccharides (LPS) of the outer membrane component trigger inflammatory cascades with life-threatening infections [5]. Unsurprisingly, *Klebsiella* infection is one of the important causes of sepsis, a syndrome of imbalanced responses to pathogens [6,7], which is an important worldwide health-care problem [8]. Interestingly, the effective iron acquisition, nitrogen source utilization, and capsules of *K. pneumoniae* lead to tolerance against harsh micro-environments resulting in the detection of *K. pneumoniae* in several conditions of gut dysbiosis (a selected growth of specific groups of organisms causing an imbalance of gut microbiota that is associated with an unhealthy outcome) [9,10]. Within the harsh microenvironment, there is competition among several groups of organisms, and *K. pneumoniae* is frequently found as the surviving organismal group [9,10].

However, Gram-negative bacteria are not the only gut organisms; gut fungi (especially *Candida albicans*) are the second most abundant organisms in the human gut that can also be a cause of nosocomial infection [11] with a high mortality rate [12]. During gut dysbiosis in critical patients (from antibiotics, reduced blood perfusion, and hypoxic conditions), *C. albicans* is one of the predominant species that are isolated from the gut, and the isolation of *Candida* is frequently associated with the colonization of high virulence *K. pneumoniae* (such as the Carbapenemase-producing bacteria) [11]. Unsurprisingly, “*Candida* colonization index” is used for the monitoring of *Candida* infection in sepsis and the disease severity of systemic infection [13]. In sepsis, the enterocyte injury from gut dysbiosis and other sepsis factors causes the translocation of microbial molecules, such as LPS and (1→3)-β-D-glucan (BG), the major cell wall component of Gram-negative bacteria and *Candida*, respectively, from the gut into the blood circulation (gut translocation) that aggravates sepsis severity [14,15]. Despite the numerous organisms in the gut, the pathogen molecules are kept inside the gut and protected from blood circulation by the gut barrier; the pathogenic organisms cannot cause active infection in the healthy gut [14]. However, gut barrier defects can be demonstrated in several situations (with and without diarrhea) such as with drugs [10,16], sepsis [15,17], obesity [18], and heavy exercise [19]. In mouse models, low-dose dextran sulfate solution (DSS) has been used to induce a chronic gut barrier defect as the onset of diarrhea is delayed after the administration [20,21]; this model might be suitable to test the chronic presence of *K. pneumoniae* and *C. albicans* against the host responses. 

Because (i) both *K. pneumoniae* and *C. albicans* can be detected in the gut during dysbiosis [10,12], (ii) the co-presence of both organisms in sepsis is possible [11,22], and (iii) the impact of *C. albicans* on *K. pneumoniae* in the intestines during gut mucosal barrier defect is still unknown, tests of the chronic co-presence of both organisms are interesting. Hence, we investigate the pathogenesis of *C. albicans* and *K. pneumoniae* in synergy in chronic low dose DSS-induced colitis mice.

## 2. Results

### 2.1. Candida Administration Worsened Disease Severity in K. pneumoniae-Administered DSS Mice Partly through Gut Barrier Defect and Systemic Inflammation 

In healthy non-DSS mice, the administration of either *Candida* or *Klebsiella* did not affect body weight, diarrhea, or mortality (Figure 1A–C) supporting the presence of the healthy gut barrier. With DSS, *Candida* administration alone did not worsen the disease severity compared with DSS alone (data not shown), while *Klebsiella* and *Klebsiella* plus *Candida* (DSS+Kleb+Can) induced higher severity as determined by weight loss, stool consistency index (an earlier loose stool), mortality, gut barrier defect (FITC-dextran assay, bacteremia, endotoxemia, and serum BG) (most severe in the DSS+Kleb+Can group) (Figure 1A–G). Notably, the presence of fluorescein isothiocyanate (FITC)-dextran (a gut un-absorbable carbohydrate) in blood at 3 h after an oral administration and the spontaneous detection of microbial molecules (viable bacteria, LPS, and BG) in the blood indicate a gut barrier defect (gut leakage) in the model [23]. The diarrheal symptoms (loose stool) in DSS+Kleb+Can mice developed as early as 2 weeks of the experiments, while for other groups, diarrhea started after 4 weeks (Figure 1B), implying a higher enterocyte vulnerability to the injury. While the mortality for DSS alone and DSS with *Klebsiella* or *Candida* was not significantly different to the control, DSS+Kleb+Can showed the highest mortality (Figure 1C). 

With DSS administration, the presence of the pathogen and organismal molecules increased serum cytokines (IL-6, but not TNF-α, and IL-10) (Figure 1H–J) together with the cytokine levels in several organs (intestines, livers, and kidneys); this was most prominent in DSS+Kleb+Can mice (Figure 2A–C). The local and systemic inflammation as indicated by cytokines in organs and in serum, respectively, were not demonstrated in the mice without DSS, despite the administration of *Candida* or *Klebsiella* (Figure 2A–C). These data also support the protective effect of a healthy gut barrier. Additionally, mice in the DSS+Kleb+Can group significantly enhanced intestinal injury as indicated by some of pro-inflammatory and anti-inflammatory cytokine levels (IL-6, TNF-α, and IL-10) in the caecum and colon as compared to DSS+Kleb and DSS-uninfected groups (Figure 2A–C). The cytokines in kidneys, livers, and large bowels (caecum and colon) of DSS+Kleb+Can group were higher than in other groups (the highest levels were in kidneys) (Figure 2A–C). The liver injury as indicated by liver histology (especially inflammatory cells infiltration) and liver apoptosis (activated caspase-3 staining) in *Klebsiella*-administered DSS or DSS+Kleb+Can mice were higher than in other groups (Figure 2D,E and Figure 3A,B); this supports the possible direct delivery of pathogenic molecules to the liver through the portal veins [14]. Notably, the upregulation of TNF-α in the intestines and kidneys but not in serum suggested an impact of local intestinal inflammation and the delivery of pathogen molecules to kidneys (possibly for excretion), respectively, which are, perhaps, not severe enough to induce TNF-α production from the circulatory immune cells. There was no liver injury and apoptosis in non-DSS mice. The liver damage in DSS alone was lower than in *Klebsiella*-administered DSS or DSS+Kleb+Can mice (Figure 2D and Figure 3A). Liver apoptosis in DSS+Kleb+Can mice was higher than in the *Klebsiella*-administered DSS group, despite a non-different injury score from H&E staining (Figure 2E and Figure 3B). 

### 2.2. Candida Slightly Altered Gut Dysbiosis in Klebsiella-Administered DSS Mice 

The increased *Candida* in the gut might select some groups of bacteria, such as glucanase-positive bacteria (bacteria that can utilize glucan molecules of fungal cell walls as the bacterial nutrients [24,25]). Hence, fecal microbiome analysis was performed. Without DSS, *Klebsiella* or *Candida* administration demonstrated gut dysbiosis as indicated by the decrease and increase in Bacteroides and Firmicutes, respectively, in comparison with controls without an alteration in bacterial diversity (Chao1 and Shannon index) (Figure 4A–E and Figure 5A–G). Additionally, the non-metric multidimensional scaling (NMDS; a non-constrained data dimensionality reduction analysis method based on Thetayc dissimilarity) demonstrated a similarity within the group of non-DSS control and *Klebsiella* (Figure 5H), implying a difference of gut bacteria among these groups. The fecal abundance of Proteobacteria (the pathogenic bacteria) [20,26] and Actinobacteria (mostly Gram-positive bacteria that produce bioactive anti-biofilm compounds [27]) in non-DSS *Candida* mice were higher than in the non-DSS control group (Figure 5C,E). The increased Actinobacteria in gut might be one of the components of the dysbiosis. In DSS-administered mice without organismal gavage, there was a decrease and increase in Bacteroidetes and Actinobacteria, respectively, when compared with non-DSS controls; however, the administration of *Klebsiella* or *Candida* in DSS mice did not alter gut bacteria, except for the highest Deferribacteres (the Gram-negative fermenting bacteria [28]) in DSS+Kleb+Can group (Figure 5A–F). The increased *Mucispirillum* sp. (the mucin degrader bacteria in phylum Deferribacteres) in DSS+Kleb+Can mice (Figure 5G) might worsen intestinal integrity [29]. In comparison with non-DSS control, 3 months of *Klebsiella*, *Candida*, or DSS administration altered some components of fecal bacteria (Figure 5A–G). Although the additional gavage of *Klebsiella* in DSS did not alter the abundance of most fecal bacteria when compared with DSS alone, the NMDS demonstrated the separation between DSS alone and DSS with *Klebsiella* (Figure 5G) indicating a change in gut microbiota. Likewise, *Candida* gavage in DSS+*Klebsiella* mice mildly affected gut microbiota when compared with DSS+*Klebsiella* mice, as indicated by the increased Deferribacteres, *Mucispirillum*, and NMDS analysis (Figure 5F–H). Hence, the profound severe sepsis in DSS+Kleb+Can compared with DSS+*Klebsiella* mice might be due to the increase in Deferribacteres with a direct impact of *Candida* or BG (a major molecule of the fungal cell wall) on several organs.

### 2.3. Molecules of Candida Induced Inflammation against Enterocytes, Hepatocytes, and Macrophages 

Because of (i) the direct contact of *Candida* with enterocytes [16,20] and (ii) the possible activation of hepatocytes and macrophages through gut translocation of *Candida* molecules to liver through blood circulation [14,18], in vitro experiments using the heat-killed preparations were performed. In enterocytes, all activations (*Klebsiella* plus *Candida* or each component alone) induced supernatant IL-8 from Caco-2 cells at 24 h of incubation (most prominent levels in Kleb+Can following by Kleb and Can alone), while the inflammatory genes (*IL-8*, *TNF-α,* and *TLR-2*) were most prominently upregulated at 4 h of incubation in the Kleb+Can group (non-upregulation in Kleb or Can alone when compared with controls) (Figure 6A–D). At 4 h post-incubation, Kleb+Can also demonstrated enterocyte damage through the most profound downregulated *Occludin* (tight junction) and reduced enterocyte integrity (transepithelial electrical resistance; TEER) without an alteration in the expression of *Muc2* gene (mucus production) (Figure 6E–G). However, the reduced TEER of Kleb or Can alone was not different to the Kleb+Can group (Figure 6F,G). 

In hepatocytes (HepG2), Kleb+Can induced higher supernatant TNF-α (but not IL-6 and IL-10) than Kleb alone at 72 h post-incubation, with the upregulation of genes for inflammation and apoptosis (*TNF-α*, *Casp8* and *Casp9*, but not *IL-6* and *bcl-2*) at 4 h post-incubation (Figure 7A–I). The downregulation of *IL-10* (a gene of anti-inflammatory cytokine) in Kleb+Can when compared with Kleb or Can alone (Figure 7F) indicated an additive effect of *Candida* on *Klebsiella*-induced inflammation in hepatocytes. Likewise, Kleb and Kleb+Can, but not Can alone, induced macrophage inflammation (supernatant IL-6, TNF-α, and IL-10) and inflammatory genes (*IL-6*, *TNF-a*, *IL-10*, *IL-8*, and *NF-κB*) at 4 h and 2 h post-incubation, respectively, when compared with controls (Figure 8A–H). Meanwhile, *NOS2* (another pro-inflammatory gene) was upregulated by all activations (Kleb, Can, and Kleb+Can) (Figure 8I). The synergy of *Candida* on *Klebsiella*-induced inflammation in THP-1-derived macrophages was indicated through profound upregulated IL-6 in Kleb+Can compared with Kleb (non-upregulation in Can alone when compared with controls) (Figure 8D). Hence, the synergy of *Candida* molecules on pro-inflammation of enterocytes, hepatocytes, and macrophages was demonstrated by some parameters through the in vitro experiments. 

## 3. Discussion

### 3.1. Candida Enhanced Inflammation in Klebsiella-Administered DSS Mice Mainly through the Activations by Microbial Molecules Rather than Gut Dysbiosis 

*Candida* abundance in mouse guts is less than in the human intestines as *C. albicans* in mouse feces can be detected by PCR [30], but not by culture [31], which is different from human conditions [32]. Gut bacterial dysbiosis by the presence of intestinal *Candida* has been shown [20,33]. Additionally, the increase in *Candida* colonization in gut is mentioned in several conditions such as antibiotic use, intestinal inflammation, DSS administration, and bacterial dysbiosis itself [34,35,36]. Thus, gut *Candida* might induce bacterial dysbiosis and gut inflammation from bacterial dysbiosis might also facilitate gut *Candida* colonization, constituting a vicious cycle. In comparison with controls, 3 months administration of *Klebsiella* or *Candida* altered gut microbiome analyses (decreased Bacteroides and increased Actinobacteria) but not induced intestinal inflammation, as indicated by the normal gut barrier (FITC-dextran, endotoxemia, and serum BG) and intestinal cytokines when compared with the control. These data imply impacts of the protective properties of the healthy gut barrier (mucus, antimicrobial peptides, intestinal immune responses) in the mice [37,38,39]. Despite the administration of *Klebsiella*, the abundance of Proteobacteria and Enterobacteriaceae (the phylum and the family of genus *Klebsiella*) in *Klebsiella*-administered mice did not rise higher than in controls suggest that the administered-*Klebsiella* might be non-sustainable. Perhaps, the 24 h duration between the last dose of microbial gavage to the time of fecal collection might be enough to clear most of the organisms from the feces. Similarly, the orally administered bacteria are non-detectable within 72 h even after the 1-month gavage of probiotics in mice [40]. Meanwhile, *Candida* gavage increased Proteobacteria possibly due to the selection of some gut bacteria that are able to utilize the *Candida* [26]. 

With 3 months of low dose DSS (without organisms), the intestinal inflammation was severe enough to retard weight gain but did not cause weight loss nor increased mortality (some mice die in this group but was not significantly different from the control). However, the administration of *Klebsiella* or *Klebsiella* plus *Candida* in DSS mice worsened the mortality. Hence, the prominent injury in the gut of DSS+Kleb+Can mice might be due to the inflammation from the daily microbial administration with the increased Deferribacteria [29,41]. As such, the increased *Mucispirillum* sp. (the putative mucin degrader in phylum Deferribacteres) in DSS+Kleb+Can mice might worsen the intestinal integrity in this group and cause a more severe gut translocation of microbial molecules [29]. The increased Deferribacteria (Gram-negative bacteria) might also elevate LPS in the gut contents and facilitate more prominent gut translocation of LPS that induces higher inflammatory responses. 

### 3.2. Candida Enhanced Hyper-Inflammation in the Intestines, Livers, and Macrophages of Klebsiella-Administered DSS Mice 

The synergistic hyper-inflammatory activation of bacteria and *Candida* have been frequently mentioned through promotion of inflammation [15,16,17,20] or adherence of bacteria to mucosa [42]. Additionally, *K. pneumoniae* is a highly virulent organism that effectively penetrates across the gut through several mechanisms (such as Rho GTPase- and Phasphatidylinositol 3-kinase/Akt pathways [43]) both transcellular (intracellular) and the paracellular (intercellular) routes [44,45,46]. Here, *Klebsiella* plus *Candida* in DSS mice showed the most prominent inflammation in the gut (especially in colons), livers, kidneys, and systemic responses. With the close proximity of the administered organisms with intestinal cells, the preparations from *Klebsiella* plus *Candida* more prominently induced enterocytes (especially with IL-8 and TNF-α production), partly through TLR-2 (one of the receptors against molecules of both organisms [47]), which might be responsible for the profound gut barrier defect and intestinal inflammatory cytokines in DSS+Kleb+Can mice. Although the administration of *Klebsiella pneumoniae* and *Candida albicans* might not cause colonization in the mouse gut, they possibly increase LPS and BG (the main cell wall components of *Klebsiella* and *Candida*, respectively). 

Subsequently, the profound gut leakage induced the prominent translocation of LPS and BG from the gut into the blood circulation, as indicated by the highest levels of LPS and BG in serum of DSS+Kleb+Can mice. Because of the direct delivery of molecules from gut translocation to the liver through the portal vein [14], hepatocytes were profoundly activated as liver apoptosis was most prominent in DSS+Kleb+Can mice. Not only the profound direct inflammation from the preparations of *Klebsiella* plus *Candida* compared with *Klebsiella* alone acted on the HepG2 hepatocytes, but also the higher abundance of LPS and BG in serum of DSS+Kleb+Can mice compared with DSS+Kleb mice might be responsible for the more prominent liver injury in the combined bacterial-fungal activations. The impact of fungal activation on hepatocytes was supported by upregulation of both *Casp8* and *Casp9* (apoptosis molecules) in HepG2 cells by the preparations from *Candida* alone. These data also support i) an impact of gut translocation of *Candida* molecules in liver injury [23,48] and ii) the apoptosis facilitation by *Candida* [49,50,51]. Likewise, the profound systemic pro-inflammation on DSS+Kleb+Can compared with DSS+Kleb mice might be mainly due to the higher LPS and BG in the former group with the synergy on macrophage responses of *Candida*-bacterial preparations. Despite frequent demonstration of the profound inflammatory synergy of BG on LPS-induced inflammation in several cells [18], the synergy using the crude preparations of fungi and bacteria was not that prominent, perhaps due to a requirement for some particular ratio between the molecules. Hence, the co-presence of *Klebsiella* and *Candida* in the gut induced a prominent enterocyte inflammation with profound gut barrier defects and gut translocation of microbial molecules (bacteria and fungi) that subsequently caused hyper-immune responses (Figure 9). Although *Candida* and *Klebsiella* were administered together in our experiments, the administration of *Candida* after the pre-formed *Klebsiella* colonization in the gut of DSS mice might also exacerbate sepsis severity similar to the simultaneous *Candida-Klebsiella* colonization.

For the clinical translation, a monitoring of intestinal fungal abundance, especially in patients with *Klebsiella* colonization in the gut, might be beneficial as they are vulnerable to the hyper-inflammation from gut barrier damage. Indeed, the enhanced colonization of fungi and *Klebsilla* spp. was reported in patients with inflammatory bowel disease (IBD) [52,53] and manipulation of *Klebsiella* and/or *Candida* in the gut of these patients might be beneficial. In IBD treatment, there are debates on the use of antibiotics and/or microbiota alteration (probiotics and fecal microbiota) that affect both fungi and bacteria [54,55,56]. The benefits of these interventions might depend on the organisms that predominantly colonize each patient. Additionally, co-infection between *Candida* and *Klebsiella* in patients with sepsis is reported, suggesting the inter-kingdom association between these organisms may have several mechanisms, including biofilm formation [22,57]. Moreover, some sepsis treatment strategies might be beneficial in sepsis from *Candida* and *Klebsiella* as the increased serum IL-6 in our mice might be attenuated by the interference of IL-6 and/or Janus kinase-signal transducer and activator of transcription (JAK/STAT) pathway [58,59]. Notably, the synergistic pro-inflammatory effect in mice after intraperitoneal injection of *Candida* and *Klebsiella* might be another similar model to test the inter-kingdom relationship between these organisms. More clinical studies on this topic would be of interest. 

## 4. Materials and Methods

### 4.1. Animals and Animal Models 

The experimental protocol in accordance with the US National Institutes of Health standards (NIH publication No. 85-23, revised 1985) was approved by the Institutional Animal Care and Use Committee of the Faculty of Medicine, Chulalongkorn University, Bangkok, Thailand. Male, 8-week-old C57BL/6 mice were purchased from the Nomura Siam International Co., Ltd. (Lumphini, Pathumwan, Bangkok, Thailand). Then, dextran sulfate sodium (DSS) was used to induce gut barrier damage using 0.8% (*w*/*v*) DSS (Sigma-Aldrich, St. Louis, MO, USA) diluted into the drinking water at the same time of bacterial-fungal administration. Mice were randomly divided into 6 groups, including water control with daily gavage by *Klebsiella pneumoniae* (Kleb) or *Candida albicans* (Can) or DSS alone or DSS with daily gavage by *K. pneumoniae* (DSS+Kleb) or *Candida albicans* (DSS+Can) or both organisms (DSS+Kleb+Can). As such, *K. pneumoniae* (ATCC 13883) from the American Type Culture Collection (ATCC, Manassas, VA, USA) was cultured on Tryptic soy agar (TSA) (Oxoid Ltd., Hampshire, UK) at 37 °C for 24 h under aerobic conditions before being quantitatively prepared using an optical density of 600 nanometers (OD600) in 0.25 mL phosphate buffer solution (PBS). In parallel, *C. albicans* (ATCC 90028) was cultured on Sabouraud dextrose broth (SDB) (Oxoid Ltd., Hampshire, UK) at 35 °C for 24 h before enumeration by hemocytometer and preparation in 0.25 mL PBS. Then, 0.25 mL of *K. pneumoniae* (1 × 10^9^ CFU) or *C. albicans* (1 × 10^6^ CFU) at 8:00 AM followed by 0.25 mL PBS at 9:00 AM was daily orally administered for 3 months for the non-combined gavage, while *K. pneumoniae* (1 × 10^9^ CFU) at 8:00 AM followed by 0.25 mL *C. albicans* (1 × 10^6^ CFU) at 9:00 AM were administered for the bacterial-fungal combination. Mice were observed daily and sacrificed (cardiac puncture) at 12 weeks, with sample collection using isoflurane anesthesia. The organs, including ileum (proximal to caecum), caecum, colon (distal to caecum), livers, and kidneys were kept in 10% neutral formalin or snap-frozen in liquid nitrogen (followed by −80 °C storage) for histology and tissue cytokines, respectively. The stool consistency was semi-quantitatively evaluated according to the following score; 0, normal; 1, soft or loose; 2, diarrhea, as previously published [60].

#### 4.1.1. Gut Permeability Determination

Gut permeability was determined by fluorescein isothiocyanate-dextran dextran (FITC-dextran) assay and the spontaneous detection of blood bacteria (culture) or microbial molecules, including lipopolysaccharide (LPS or also known as endotoxin) and (1→3)-β-D-glucan (BG) in serum, as previously described [10,23]. Positive viable bacteria or microbial molecules in blood without active infection or the detection of FITC-dextran (a non-intestinal absorbable carbohydrate) in serum after an oral gavage indicates a gut permeability defect [9,10]. For the FITC-dextran assay, FITC-dextran (FITC-dextran; molecular weight 4.4 kDa) (Sigma-Aldrich, St. Louis, MO, USA) at 25 mg/mL (0.5 mL) was orally administered for 3 h before sacrifice and serum FITC-dextran was measured with fluorospectrometry (Thermo Scientific, Wilmington, DE, USA) with the excitation and emission wavelength at 485 and 523 nm, respectively, using a standard curve of serially diluted FITC-dextran [9,10]. For bacteremia, 20 μL of blood was directly spread onto blood agar plates (Oxoid Ltd., Germany) and incubated at 37 °C for 24 h before enumeration of bacterial colonies. In parallel, LPS and BG in serum were measured by HEK-Blue LPS detection (InvivoGen, San Diego, CA, USA) and Fungitell (Associates of Cape Cod, Falmouth, MA, USA). The values of LPS < 0.01 EU/mL and BG < 7.8 pg/mL were recorded as 0 due to the limitation of the standard curves.

#### 4.1.2. Serum and Histological Analyses 

The cytokines, including tumor necrosis factor (TNF)-α, interleukin (IL)-6, and IL-10, in serum and in homogenized tissue, were determined by an enzyme-linked immunosorbent assay (ELISA) (Invitrogen, Carlsbad, CA, USA). The tissues were homogenized by the Ultra-Turrax homogenizer (IKA, Staufen, Germany) in 500 µL of PBS (pH 7.4) with protease inhibitors and centrifuged at 12,000× *g* for 15 min at 4 °C to separate the supernatant. The cytokine in the supernatant represented tissue inflammation [16,20]. For histology, the samples were fixed in 10% formalin, embedded in paraffin, cut to a 4 μm thickness, and stained with Hematoxylin-Eosin (H&E) color. Liver injury was evaluated, based on cell congestion, cellular degeneration, cytoplasmic vacuolization, leukocyte infiltration, and cellular necrosis, in 10 randomly selected fields at ×200 magnification for each animal with the following score per examined field: 0 indicates an area of damage less than 10%; 1 indicates damage 10–25%; 2 indicates damage involving 25–50%; 3 indicates damage involving 50–75%, and 4 indicates 75–100% of the area being affected; this was the scoring used in a previous publication, with modification [61]. In parallel, hepatic apoptosis was evaluated by immunohistochemistry with an anti-active caspase-3 antibody (cell signaling). The apoptotic cells per slide were counted at 200× magnification and expressed as positive cells per high-power field as previously published [62]. 

#### 4.1.3. Fecal Microbiome Analysis

The fecal microbiota analysis was performed as previously reported with minor modifications [20,63]. In short, feces (0.25 g/mice) were extracted for metagenomic DNA with the DNeasy PowerSoil Kit (Qiagen GmbH, Hilden, Germany) and qualified by agarose gel electrophoresis with nanodrop spectrophotometry. Each DNA sample was subjected to 16S rRNA gene V3-V4 region amplification using universal primers 341F (forward; 5′-CCTACGGGNGGCWGCAG-3′) and 805R (reverse; 5′-GACTACHVGGGTATCTAATCC-3′), in 2× SparQ HiFi PCR Master Mix (QuantaBio, Beverly, MA, USA). The thermocycling condition included an initial denaturation step at 98 °C for 2 min, followed by 30 cycles of 98 °C 20 s, 60 °C 30 s, and 72 °C 60 s, and then a final extension step at 72 °C 1 min. Subsequently, each 16S rRNA amplicon was purified using SparQ Puremag Beads (QuantaBio, Beverly, MA, USA), and indexed using 5 µL of Nextera TX index primer by additional 8-10 cycles of PCR condition above. The final products were cleaned, pooled, and diluted to the final loading concentration at pM. Sequencing was performed by 2 × 250 bp pair-end sequencing following Illumina MiSeq protocols (Illumina, San Diego, CA, USA) [64]. Sequences were analyzed with Mothur version 1.39.1’s standard operating procedures for MiSeq [65]. 

### 4.2. The In Vitro Experiments

The inflammatory responses against the media control or the molecules from *Klebsiella* or *Candida* or *Klebsiella* plus *Candida* (using heat-killed preparations) were tested in several cell lines, including human colonic epithelial Caco-2 cells (ATCC HTB-37), human hepatoma HepG2 cells (ATCC HB-8065), and the macrophages derived from human monocytoid THP-1 cells (ATCC TIB- 202). Then, *K. pneumoniae* (1.5 × 10^9^ cells/mL) [10] or *C. albicans* (5 × 10^7^ cells/mL) [16,66] were heated (65 °C for 1 h), sonicated (power amplitude 40%, pulse on and off for 20 and seconds, respectively, for 60 min) using a High-Intensity Ultrasonic Processor (VC/VCX 130, 500, 750). The multiplicity of infection ratios (MOI) for *K. pneumoniae* or *C. albicans* (the ratio between numbers of the cells and the organisms) were 1:300 and 1:10, respectively, following previous publications [10,20]. In parallel, each cell line was differently maintained following the previous protocols [10,18]. Briefly, Caco-2 and HepG2 were maintained in Dulbecco’s modified Eagle medium (DMEM) supplemented with 20% and 10% (*v*/*v*) heat- inactivated fetal bovine serum (Gibco-Invitrogen, Grand Island, NY, USA), respectively, while THP-1 were maintained in RPMI 1640 medium supplemented with 10% heat-inactivated fetal bovine serum (Gibco-Invitrogen) at 37 °C in a humidified 5% CO_2_. For macrophage differentiation, THP-1 were incubated with phorbol 12-myristate 13-acetate (PMA) (Sigma-Aldrich) 100 nM for 3 days, washed with RPMI 1640 and Hank’s Balanced Salt Solution (HBSS) (Gibco-Invitrogen) for 30 min [66]. Then, Caco-2 or HepG2 at 5 × 10^4^ cells/well or THP-1-derived macrophages at 1 × 10^5^ cells/well were incubated with media alone or media with the preparations from *Klebsiella* or *Candida* or *Klebsiella* plus *Candida* (with the same volume per well) before collection of supernatant and cells at different time-points. 

The supernatant was collected by centrifugation (125× *g*, 4 °C for 7 min) and cytokines were measured by ELISA kit (Invitrogen, Carlsbad, CA, USA) according to the manufacturer’s instructions. In parallel, the gene expression was measured by quantitative reverse-transcription polymerase chain reaction (qRT-PCR) as previously described [10]. In brief, total RNA was extracted from the treated cells using TRIzol reagent (Invitrogen, USA). The RNA (50 ng) was converted into the complementary DNA (cDNA) by high capacity reverse transcription assay (Applied Biosystems, Warrington, UK) and SYBR Green PCR Master Mix using a QuantStudio™ Design & Analysis Software v1.4.3 (Thermo Fisher Scientific, Foster City, CA, USA) in relation to *GAPDH* expression using 2^−ΔΔCp^ method [67]. The list of primers for PCR is presented in Table 1.

In addition, enterocyte integrity was evaluated by transepithelial electrical resistance (TEER) as previously published [40,63]. In short, Caco-2 cells at 5 × 10^4^ cells/well were seeded onto the upper compartment of 24-welled Boyden chamber Transwell using supplemented DMEM under 5% CO_2_ at 37 °C for 15 days with daily media replacement to establish the confluent monolayer. The cells were treated with the heat-killed preparation from *K. pneumoniae* at the multiplicity of infection (MOI) 1:300 either alone or with heat-killed *C. albicans* (MOI 1:10) in 5% CO_2_ at 37 °C before TEER measurement by an EMOM^2^ Epithelial Voltohmmeter with a chopstick electrode (World Precision Instruments, Inc., Sarasota, FL, USA). The value of TEER was ohm (Ω) × cm^2^.

### 4.3. Statistical Analysis 

Mean ± standard error of the mean (SEM) was used for data presentation. The differences between groups were examined by one-way analysis of variance (ANOVA) followed by Tukey’s analysis or Student’s *t*-test for comparisons of multiple groups or 2 groups, respectively. Survival analysis was performed by Log-rank test. All statistical analyses were performed with Graph Pad Prism version 9.0 software (La Jolla, CA, USA). A *p*-value of <0.05 was considered statistically significant.

## 5. Conclusions

An impact of fungi in the gut enhancing sepsis severity was demonstrated by the 3 months co-administration of *Klebsiella* and *Candida* in a dextran sulfate-induced colitis mouse model. The presence of intestinal *Candida* directly facilitated *Klebsiella*-induced enterocyte injury and indirectly increased some pathogenic bacteria causing a prominent gut barrier damage and profound gut translocation of microbial molecules that worsen sepsis severity. The individuals with the co-presence of *Klebsiella* and *Candida* might be monitored.

## Figures and Tables

**Figure 1 ijms-23-07050-f001:**
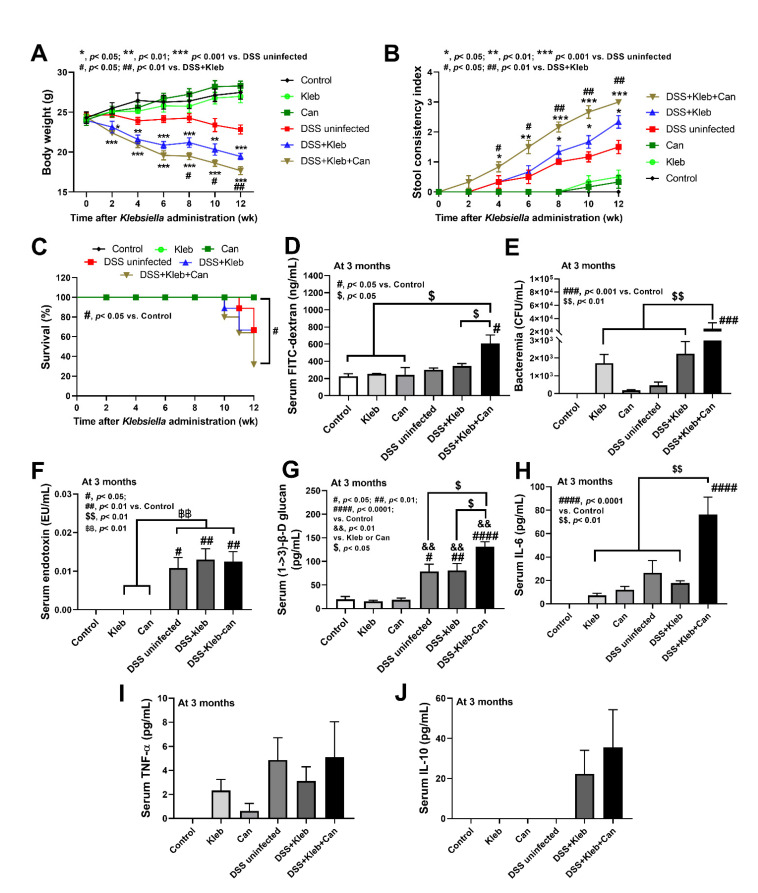
The characteristics of mice after daily gavage with phosphate buffer solution (control), *K. pneumoniae* (Kleb), or *C. albicans* (Can) with or without dextran sulfate solution (DSS), as indicated by body weight (**A**), stool consistency index (**B**), survival analysis (**C**), gut barrier defect (FITC dextran assay, bacteremia, endotoxemia, and serum (1→3)-β-D-glucan (BG)) (**D**–**G**), and serum cytokines (IL-6, TNF-α, and IL-10) (**H**–**J**). (n = 8–10/time-point or group). #, *p* < 0.05; ##, *p* < 0.01; ###, *p* < 0.001; ####, *p* < 0.0001 vs. control; *, *p* < 0.05; **, *p* < 0.01; ***, *p* < 0.001 vs. DSS uninfected; $, *p* < 0.05; $$, *p* < 0.01 vs. DSS-Kleb-Can; &&, *p* < 0.01 vs. Kleb or Can; ฿฿, *p* < 0.01, the indicated groups as have determined by ANOVA with Tukey’s analysis.

**Figure 2 ijms-23-07050-f002:**
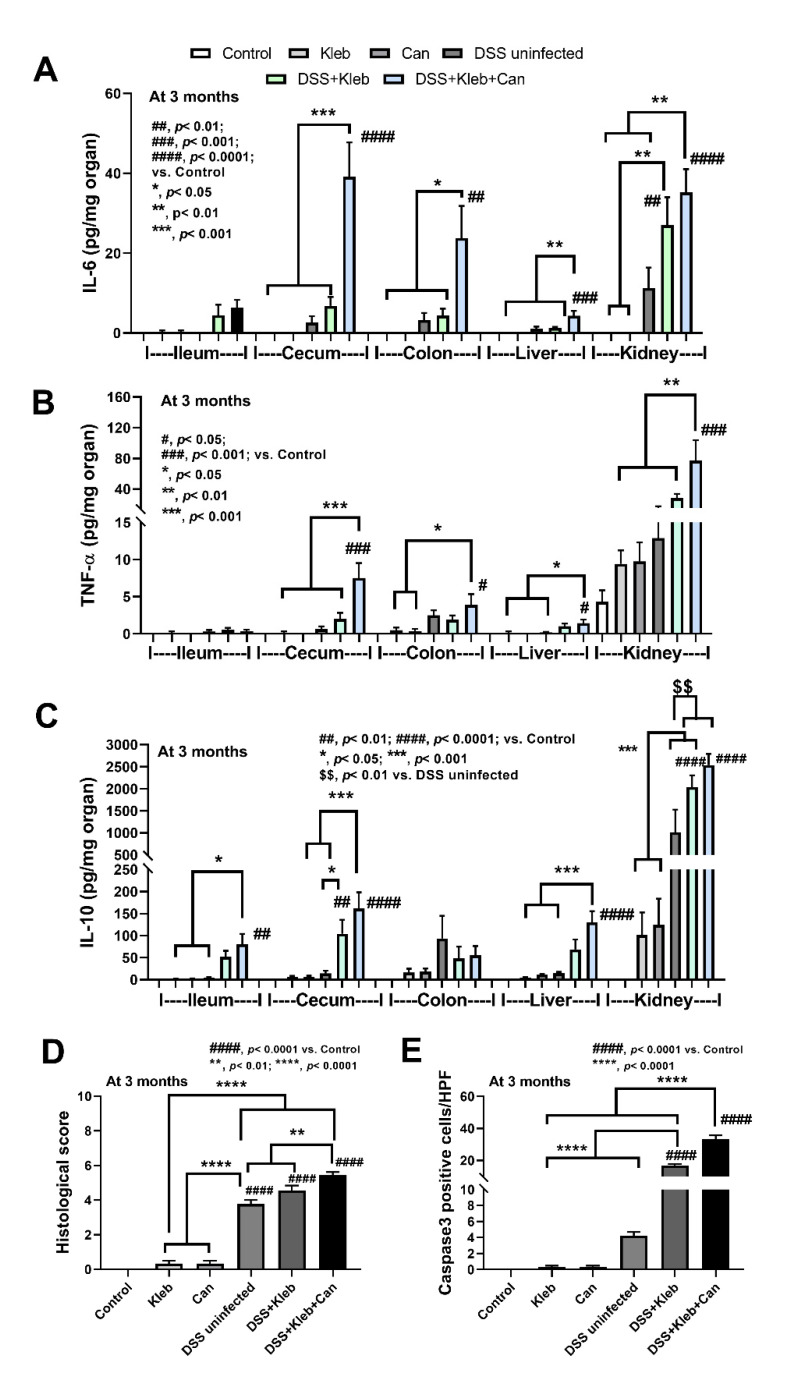
The characteristics of mice after daily gavage with phosphate buffer solution (control), *K. pneumoniae* (Kleb), or *C. albicans* (Can) with or without dextran sulfate solution (DSS) as indicated by the cytokines (IL-6, TNF-α, and IL-10) from the intestines (ileum, caecum, and colon), livers, and kidneys (**A**–**C**), liver histological scores and caspase 3 positive cell (**D**–**E**). (n = 8–10/group). #, *p* < 0.05; ##, *p* < 0.01; ###, *p* < 0.001; ####, *p* < 0.0001 vs. control; $$, *p* < 0.01 vs. DSS uninfected; *, *p* < 0.05; **, *p* < 0.01; ***, *p* < 0.001, the indicated groups as have determined by ANOVA with Tukey’s analysis.

**Figure 3 ijms-23-07050-f003:**
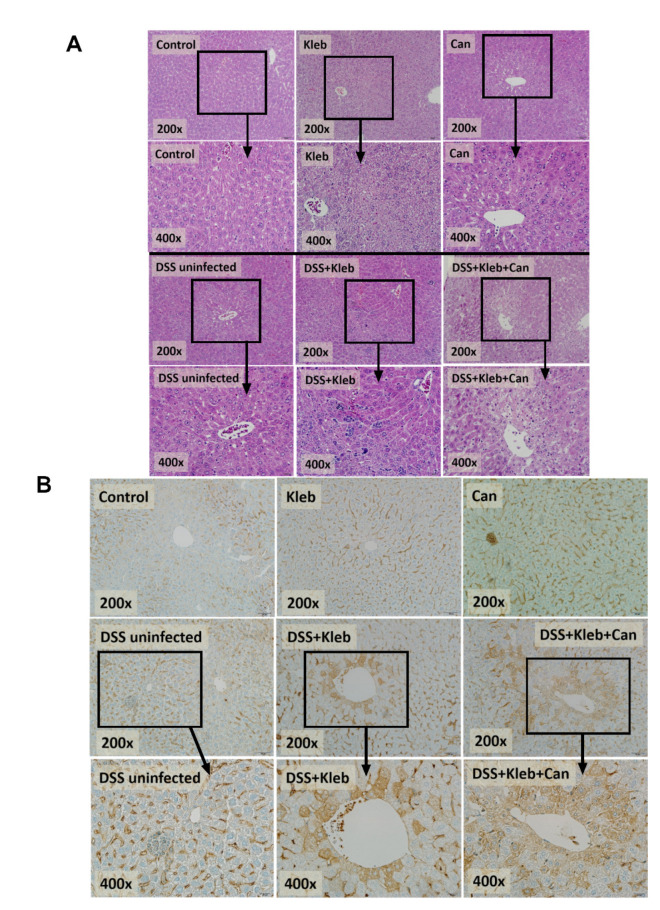
The characteristics of mice after daily gavage with phosphate buffer solution (control), *K. pneumoniae* (Kleb), or *C. albicans* (Can) with or without dextran sulfate solution (DSS) as indicated by representative pictures of liver histology using Hematoxylin & Eosin (H&E) color (**A**) and liver apoptosis (activated caspase 3 immunohistochemistry) (**B**). (n = 8–10/group) Notably, the semi-quantitative analyses of these histological pictures are presented in Figure 2D,E.

**Figure 4 ijms-23-07050-f004:**
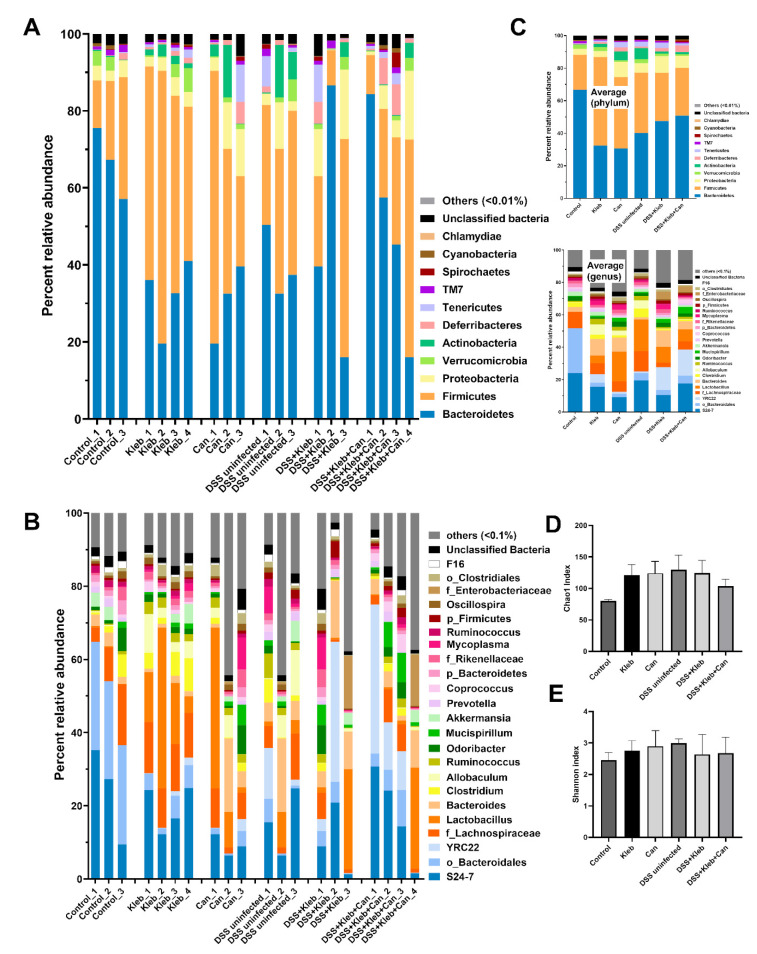
The characteristics of mice after daily gavage with phosphate buffer solution (control), *K. pneumoniae* (Kleb), or *C. albicans* (Can) with or without dextran sulfate solution (DSS) as indicated by the abundance of fecal bacteria in phylum and genus levels (**A**,**B**) with the average values (**C**), and the bacterial diversity index (Chao1 and Shannon index) (**D**,**E**). (n = 3–4/group)

**Figure 5 ijms-23-07050-f005:**
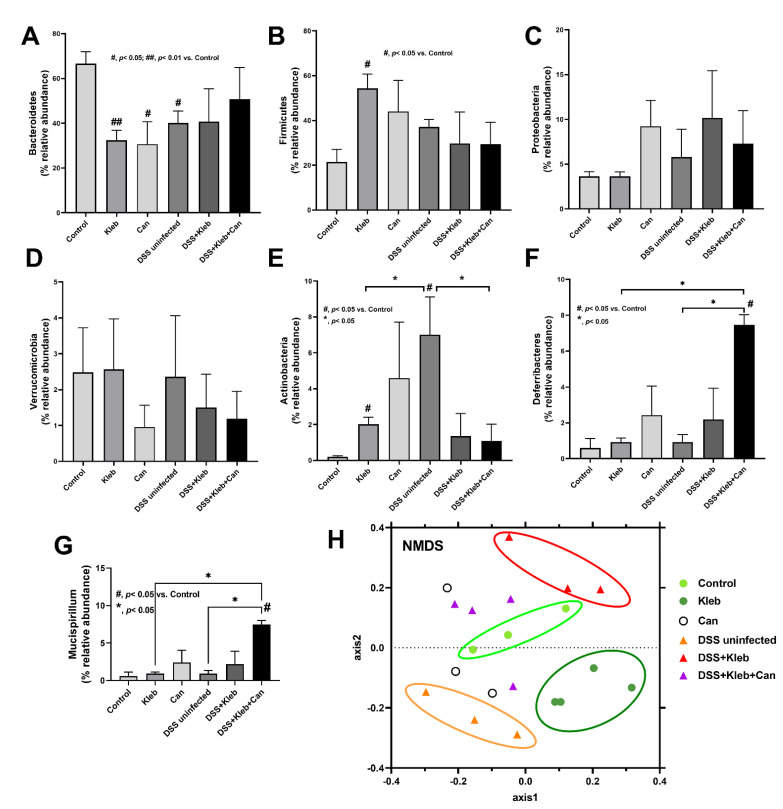
The characteristics of mice after daily gavage with phosphate buffer solution (control), *K. pneumoniae* (Kleb), or *C. albicans* (Can) with or without dextran sulfate solution (DSS) as indicated by the graph presentation of the abundance of fecal bacteria in some phylum and genus (**A**–**G**) with the Non-Metric Multidimensional Scaling (NMDS) analysis (**H**). (n = 3–4/group). *, *p* < 0.05; #, *p* < 0.05; ##, *p* < 0.01 vs. control, the indicated groups as have determined by ANOVA with Tukey’s analysis.

**Figure 6 ijms-23-07050-f006:**
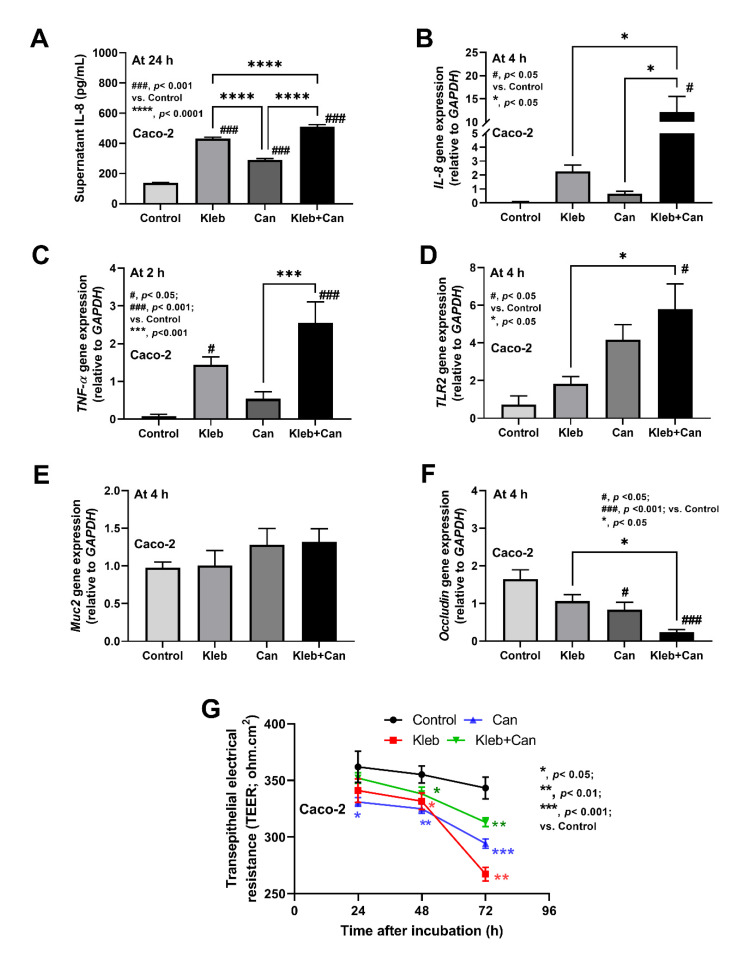
The characteristics of enterocytes (Caco-2 cells) after the activation by media control (control) and the preparations from *K. pneumoniae* (Kleb) or *C. albicans* (Can) alone or the combination (Kleb+Can) as indicated by supernatant IL-8 at 24 h post-incubation (**A**) or the gene expression (4 h post-incubation) of inflammatory molecules (*IL-8*, *TNF-α*, and toll-like receptor-2 (*TLR-2*)), mucin (*Muc2*), and tight junction (*Occludin*) (**B**–**F**) with the transepithelial electrical resistance (TEER) (**G**).(Independent triplicated experiments were performed.) #, *p* < 0.05; ###, *p* < 0.001 vs. control; *, *p* < 0.05; **, *p* < 0.01; ***, *p* < 0.001; ****, *p* < 0.0001, the indicated groups as have determined by ANOVA with Tukey’s analysis.

**Figure 7 ijms-23-07050-f007:**
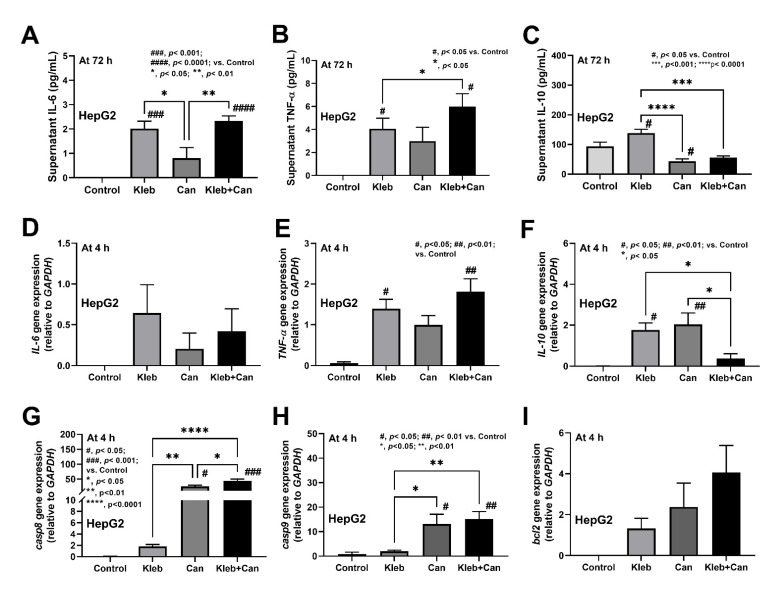
The characteristics of hepatocytes (HepG2 cells) after the activation by media control (control) and the preparations from *K. pneumoniae* (Kleb) or *C. albicans* (Can) alone or the combination (Kleb+Can) as indicated by supernatant cytokines (IL-6, TNF-α, and IL-10) at 72 h post-incubation (**A**–**C**) and the gene expression (4 h post-incubation) of the molecules of inflammation (*IL-6*, *TNF-α*, and *IL-10*) (**D**–**F**) and apoptosis (*casp8, casp9,* and *bcl-2*) (**G**–**I**). (Independent triplicated experiments were performed.) #, *p* < 0.05; ##, *p* < 0.01; ###, *p* < 0.001; ####, *p* < 0.0001 vs. control; *, *p* < 0.05; **, *p* < 0.01; ***, *p* < 0.001; ****, *p* < 0.0001, the indicated groups as have determined by ANOVA with Tukey’s analysis.

**Figure 8 ijms-23-07050-f008:**
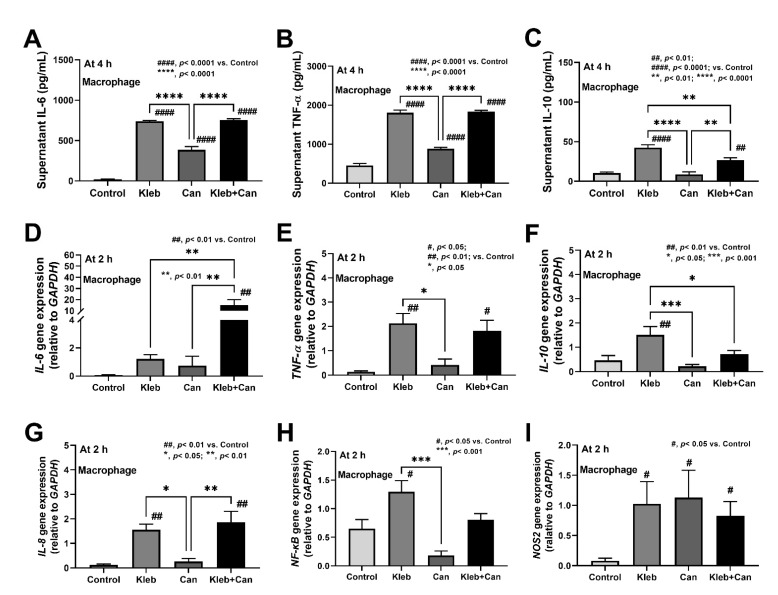
The characteristics of macrophages (THP-1-derived cells) after the activation by media control (control) and the preparations from *K. pneumoniae* (Kleb) or *C. albicans* (Can) alone or the combination (Kleb+Can) as indicated by supernatant cytokines (IL-6, TNF-α, and IL-10) at 72 h post-incubation (**A**–**C**) and the gene expression (4 h post-incubation) of the molecules of inflammation (*IL-6*, *TNF-α*, *IL-10*, *IL-8*, *NF-κB*, and *NOS2*) (**D**–**I**). (Independent triplicated experiments were performed.) #, *p* < 0.05; ##, *p* < 0.01; ####, *p* < 0.0001 vs. control; *, *p* < 0.05; **, *p* < 0.01; ***, *p* < 0.001; ****, *p* < 0.0001, the indicated groups as have determined by ANOVA with Tukey’s analysis.

**Figure 9 ijms-23-07050-f009:**
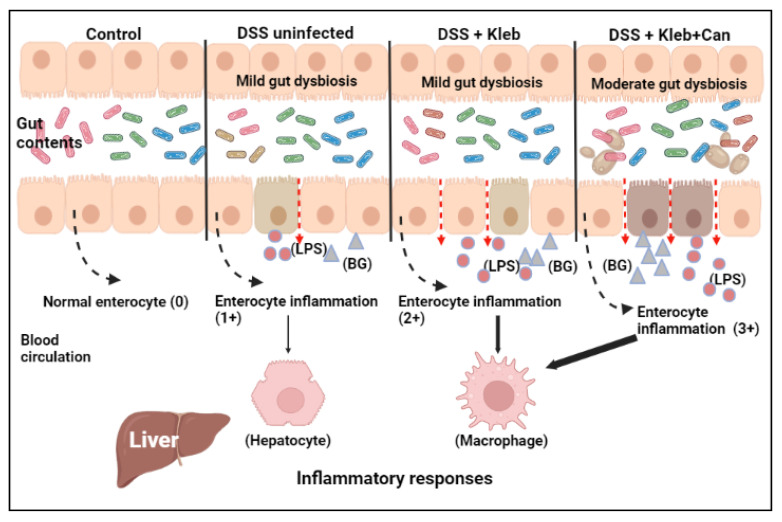
The proposed working hypothesis demonstrates the role of *Candida* with and without *Klebsiella* in dextran sulfate solution (DSS)-induced enterocyte injury. Dextran sulfate solution (DSS) with *Klebsiella* activates more enterocyte inflammation than DSS alone but is lower than the activation by DSS with *Klebsiella* plus *Candida*. The more prominent gut translocation of fungal plus bacterial molecules, as represented by lipopolysaccharide (LPS) and (1→3)-β-D-glucan (BG), in DSS with *Klebsiella* plus *Candida*-administered mice induces profound inflammation through the responses of hepatocytes and macrophages. Notably, the thickness of the solid arrows represents the intensity of the reaction due to the different abundances of microbial molecules. This picture was created by BioRender (https://app.biorender.com/). The accessed date is 27 May 2022 to create this picture.

**Table 1 ijms-23-07050-t001:** List of primers ^a^.

Target	Primer Sequence	
	Forward	Reverse
*IL-8*	5′-ACACTGCGCCAACACAGAAATTA-3′	5′-TTTGCTTGAAGTTTCACTGGCATC-3′
*TNF* *-α*	5′-CTCTTCTGCCTGCTGCACTTTG-3′	5′-ATGGGCTACAGGCTTGTCACTC-3′
*IL-6*	5′-ATGAACTCCTTCTCCACAAGC-3′	5′-GTTTTCTGCCAGTGCCTCTTTG-3′
*IL-10*	5′-TCTCCGAGATGCCTTCAGCAGA-3′	5′-TCAGACAAGGCTTGGCAACCCA-3′
*NF-κB*	5′-ATGGCTTCTATGAGGCTGAG-3′	5′-GTTGTTGTTGGTCTGGATGC-3′
*Occludin*	5′-CCAATGTCGAGGAGTGGG-3′	5′-CGCTGCTGTAACGAGGCT-3′
*MUC2*	5′-CCTGCCGACACCTGCTGCAA-3′	5′-ACACCAGTAGAAGGGACAGCACCT-3′
*bcl-2*	5′-GGTGCCACCTGTGGTCCACCT-3′	5′-CTTCACTTGTGGCCCAGATAGG-3′
*Casp8*	5′-TTTCTGCCTACAGGGTCATGC-3′	5′-TGTCCAACTTTCCTTCTCCCA-3′
*Casp9*	5′-CTCAGACCAGAGATTCGCAAAC-3′	5′-GCATTTCCCCTCAAACTCTCAA-3′
*NOS2*	5′-CAGCGGGATGACTTTCCAAG-3′	5′-AGGCAAGATTTGGACCTGCA-3′
*GAPDH*	5′-GCACCGTCAAGGCTGAGAAC-3′	5′-ATGGTGGTGAAGACGCCAGT-3′

^a^ IL-8, Interleukin-8; TNF-α, Tumor necrosis factor-α; IL-6, Interleukin-6; IL-10, Interleukin-10; *NF-κB*, Nuclear factor kappa B; MUC2, mucin2; Casp8, Caspase 8; Casp9, Caspase 9; NOS2, Nitric oxide synthase 2; GAPDH, glyceraldehyde-3-phosphate dehydrogenase.

## Data Availability

The data is contained within the article.

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
