# Peer review of "Candida Worsens Klebsiella pneumoniae Induced-Sepsis in a Mouse Model with Low Dose Dextran Sulfate Solution through Gut Dysbiosis and Enhanced Inflammation"

_ijms, 2022, doi:10.3390/ijms23137050_

Round 1
Reviewer 1 Report
Excellent research.
Are all 76 references necessary for this study? Please try to cite only the most relevant 1 or 2 references for each statement.
Suggestions to include in Discussion:
Could Candida worsen existing Klebsiella-induced gut dysbiosis, for example by administration 6-8 weeks after initiation of Klebsiella in DSS-treated mice?
Do IBD patients have an increased incidence of Candida (Gut 2017 Jun;66(6):1039-1048.
doi: 10.1136/gutjnl-2015-310746) or Klebsiella infections? Could antimycotics against Candida or antibacterials against Klebsiella reduce IBD symptoms in humans?
Do these results offer a partial reason why faecal transplantation is effective in human IBD possibly by removing infectious agents such as Klebsiella and Candida?
Author Response
Are all 76 references necessary for this study? Please try to cite only the most relevant 1 or 2 references for each statement.
ANS: We thank the reviewer for the comment and reduce reference number accordingly.
Suggestions to include in Discussion:
Could Candida worsen existing Klebsiella-induced gut dysbiosis, for example by administration 6-8 weeks after initiation of Klebsiella in DSS-treated mice?
ANS: We thank the reviewer for the comment. Although we did not perform Candida administration at 6-8 wk of Klebsiella-DSS mice, we hypothesize that Candida might enhance sepsis severity in the model. We mentioned this hypothesis as a limitation of our study in the new discussion as following “Although Candida and Klebsiella were administered together in our experiments, the administration of Candida after the pre-formed Klebsiella colonization in the gut of DSS mice might also exacerbate sepsis severity similar to the simultaneous Candida-Klebsiella colonization”. Moreover, the enhanced colonization of fungi and Klebsilla spp. was reported in patients with inflammatory bowel disease (IBD) (Sokol H, Gut. 2017 Jun;66(6):1039-1048.// Atarashi K,. Science. 2017;358(6361):359-365.) and manipulation of Klebsiella and/ or Candida in the gut of these patients might be beneficial.
Do IBD patients have an increased incidence of Candida (Gut 2017 Jun;66(6):1039-1048. doi: 10.1136/gutjnl-2015-310746) or Klebsiella infections?
ANS: We thank the reviewer for the suggestion. Indeed, fungi and Klebsiella in the gut are reported in IBD. We put these informations with this reference in the new discussion as following “The enhanced colonization of fungi and Klebsilla spp. was reported in patients with inflammatory bowel disease (IBD) (Sokol H, Gut. 2017 Jun;66(6):1039-1048.// Atarashi K,. Science. 2017;358(6361):359-365.) and manipulation of Klebsiella and/ or Candida in the gut of these patients might be beneficial.”.
Could antimycotics against Candida or antibacterials against Klebsiella reduce IBD symptoms in humans? Do these results offer a partial reason why faecal transplantation is effective in human IBD possibly by removing infectious agents such as Klebsiella and Candida?
ANS: We thank the reviewer for the suggestion. Treatment of IBD through the interference of fungi by probiotics and fecal microbiota is mentioned. Then, we add in the new discussion as following “In IBD treatment, there are debates on the use of antibiotics and/ or microbiota alteration (probiotics and fecal microbiota) that affect on both fungi and bacteria (Nitzan O. World J Gastroenterol. 2016;22(3):1078-1087. // Hager CL,. Dig Liver Dis. 2017 Nov;49(11):1171-1176.// Tan P. Front Pharmacol. 2020 Sep 18;11:574533.). The benefits of these interventions might depend on the organisms that are predominantly colonized in each patient.”.

Reviewer 2 Report
Review comments of “Candida worsens Klebsiella pneumoniae induced-sepsis in a mouse model with low dose dextran sulfate solution through gut dysbiosis and enhanced inflammation” by Wimonrat Panpetch et al.
This study provided interesting observations in the effects of co-presence of K. pneumoniae and C. albicans in a mouse model. Considering that 3% to 8% of all nosocomial bacterial infections in the United States are caused by K. pneumoniae, and the Candida albicans is an abundant organism in the human gut, the effects of combined K. pneumoniae and C. albicans found in this study may pave the way for further studies in this field. Here I am providing some comments for improve:
1. Are there any data of human patients claim that Candida albicans contributes to nosocomial bacterial infections caused by K. pneumoniae? This information will help connect findings of this study directly to human infections.
2. It seems that the IL-6 upregulation is the major difference in the serum cytokines between K. pneumoniae along samples and with C. albicans samples. Previous studies claimed that IL-6 have an impact on animal body weight and lifespan. Thus, is IL-6 the cause of weight loss and survival phenotypes in Figure 1AC? Can blocking IL-6 or JAK/STAT rescue these phenotypes?
3. TNF-a is upregulated in the cecum, colon, and kidney (Figure 2B) but increased TNF-a is not detected in serum. Does that suggest TNF-a is not secreted to serum but functions only within the cell expressing it?
4. The in vitro experiments using the heat-killed preparations are very impressive, however, additional experiment needed here is to inject K. pneumoniae and C. albicans into mouse abdomen and check whether those genes are regulated in a similar manner as in those mice orally administered with them.
Minor points:
1. Figure 3, it is better to have quantifications of these figures.
Author Response
Review comments of “Candida worsens Klebsiella pneumoniae induced-sepsis in a mouse model with low dose dextran sulfate solution through gut dysbiosis and enhanced inflammation” by Wimonrat Panpetch et al.
This study provided interesting observations in the effects of co-presence of K. pneumoniae and C. albicans in a mouse model. Considering that 3% to 8% of all nosocomial bacterial infections in the United States are caused by K. pneumoniae, and the Candida albicans is an abundant organism in the human gut, the effects of combined K. pneumoniae and C. albicans found in this study may pave the way for further studies in this field. Here I am providing some comments for improve:
- Are there any data of human patients claim that Candida albicans contributes to nosocomial bacterial infections caused by K. pneumoniae? This information will help connect findings of this study directly to human infections.
ANS: We thank the reviewer for the comment. There are some reports of the co-infection between Klebsiella and Candida possibly due to the synergy on biofilm production. Then, we add this information in the new discussion as following “Moreover, the co-infection between Candida and Klebsiella in patients with sepsis is reported suggesting the interkingdom association between these organisms with several mechanisms, including the biofilm formation (Zhong L,. BMC Infect Dis. 2020 Nov 6;20(1):810.// Welp A.L., Front. Cell Infect. Microbiol. 2020;10:213.)”.
- It seems that the IL-6 upregulation is the major difference in the serum cytokines between K. pneumoniae along samples and with C. albicans samples. Previous studies claimed that IL-6 have an impact on animal body weight and lifespan. Thus, is IL-6 the cause of weight loss and survival phenotypes in Figure 1AC? Can blocking IL-6 or JAK/STAT rescue these phenotypes?
ANS: We thank the reviewer for the comment. Indeed, IL-6 is an important cytokine in sepsis that possibly responsible for the mouse characteristics and blockage of IL-6 or JAK/STAT is mentioned as sepsis attenuations. Hence, we mentioned this information in the new discussion as following “Notably, some sepsis treatment strategies might be beneficial in sepsis from Candida and Klebsiella as the increased serum IL-6 in our mice might be attenuated by the interference on IL-6 and/ or Janus kinase-signal transducer and activator of transcription (JAK/STAT) pathway (Bloomfield M,. Front Immunol. 2019 Nov 8;10:2629.// Cai B, Inflammation. 2015 Aug;38(4):1599-608. ).”.
- TNF-a is upregulated in the cecum, colon, and kidney (Figure 2B) but increased TNF-a is not detected in serum. Does that suggest TNF-a is not secreted to serum but functions only within the cell expressing it?
ANS: We thank the reviewer for the comment. Because TNF-a can be produced by nearly all cell types in human, we hypothesize that the cause of inflammation is enough to induce local inflammation but not enough to induce systemic inflammation. Then, we add this hypothesis in the new result as following “The upregulation of TNF-a in the intestines and kidneys but not in serum suggested an impact of local intestinal inflammation and the delivery of pathogen molecules to kidneys (possibly for the excretion), respectively, which are, perhaps, not severe enough to induce TNF-a production from the circulatory immune cells.”.
- The in vitro experiments using the heat-killed preparations are very impressive, however, additional experiment needed here is to inject K. pneumoniae and C. albicans into mouse abdomen and check whether those genes are regulated in a similar manner as in those mice orally administered with them.
ANS: We thank the reviewer for this interesting experiment. However, the intra-abdominal injection might accumulate in the peritoneum but not systemically absorbed that is not our primary aim of the experiment (the test of pathogen molecules in blood circulation from gut leakage). Then, we mentioned this suggestion as a further experiment in the new discussion as following “Notably, the synergistic pro-inflammatory effect in mice after intraperitoneal injection of Candida and Klebsiella might be another similar model to test the interkingdom relationship between these organisms.”.
Minor points:
- Figure 3, it is better to have quantifications of these figures.
ANS: We apologize to the unclear presentation. The scores are in the fig 2D, E. Then, we add in the fig 3 as following “Notably, the semi-quantitative analyses of these histological pictures are presented in Figures 2D and E.”.

Round 2
Reviewer 2 Report
The authors have addressed all the questions and improved the manuscript.